# Brain-Mimetic Staged Representation Learning with Disentangled Coarse and Fine Semantic for EEG Visual Decoding

## Abstract

Decoding visual information from electroencephalography (EEG) signals remains a fundamental challenge in brain–computer interfaces and medical rehabilitation. Most existing methods focus on refining EEG encoders to obtain stronger EEG embeddings for alignment with visual features, but they largely overlook that human visual perception is inherently staged, progressing from low-level feature detection to high-level semantic abstraction and ultimately to information integration. Inspired by neuroscientific theories of staged vision, we propose a novel EEG representation learning framework that explicitly models the three stages of brain visual processing: Phase-I for low-level visual representation learning, Phase-II for high-level semantic representation learning, and Phase-III for integrative information fusion. To further enhance semantic modelling, we propose (i) a multimodal dual-level semantic learning mechanism, which disentangles coarse label-level semantics and fine image-level semantics from visual EEG channels, and (ii) a new concept of virtual EEG channels, which expand the representational capacity of EEG signals. Extensive experiments on the largest benchmark dataset demonstrate significant improvements over state-of-the-art methods under both subject-dependent and subject-independent zero-shot settings, confirming both robustness and generalisability of our method. By explicitly modelling staged brain-mimetic processing and dual-level enriched semantic representations, our work not only advances decoding performance but also provides a biologically grounded perspective for future EEG-based brain decoding research.

## 1 Introduction

Decoding visual information from electroencephalography (EEG) signals is a central task in brain-computer interface research (Wilson et al., 2024; Ferrante et al., 2024b), with broad implications for neurorehabilitation (Zhang et al., 2025c), visual cognition (De La Torre-Ortiz & Ruotsalo, 2024), and brain-inspired artificial intelligence (Pereira et al., 2018; Ding et al., 2025). However, this task remains highly challenging due to the intrinsic low signal-to-noise ratio, severe nonstationarity, and complex spatiotemporal dynamics of EEG signals, which hinder the stable and accurate recovery of visual representations (Guo et al., 2025; Liu et al., 2025; Mentzelopoulos et al., 2024).

Nearly all existing methods address this challenge by refining EEG encoders to extract stronger global embeddings for alignment with visual features. Some methods emphasize noise suppression or representation enrichment, such as uncertainty-aware blur priors for suppressing noise (Wu et al., 2025), multimodal feature integration (Zhang et al., 2025b), wavelet-based contrastive learning for category-aware decoding (Zhang et al., 2025a), and diffusion-driven generative modelling (Li et al., 2024). Others extend this line of work by improving the semantic consistency of global embeddings through tailored loss functions (Chen et al., 2024), employing multimodal graph representations (Du et al., 2023), or aligning EEG embeddings with large-scale vision-language models (Ferrante et al., 2024a; Song et al., 2025). While these methods have advanced the field, they predominantly formulate EEG-based visual decoding as a global embedding alignment problem, thereby failing to capture the inherently staged nature of human visual processing. Consequently, they also neglect the potential of staged EEG embedding learning and disentangled dual-level semantic modelling.

Neuroscience provides compelling evidence that human visual perception unfolds progressively and hierarchically. Felleman & Van Essen (1991) demonstrated that the primate visual cortex follows a distributed hierarchical structure, while Goodale & Milner (1992) proposed the influential two-visual-pathway hypothesis, distinguishing the ventral ("what") and dorsal ("where/how") streams. More recent studies confirmed that category-related visual information emerges gradually in the human brain (Graumann et al., 2022), and event-related potential (ERP) research has shown that EEG signals reflect distinct cognitive functions across different phases (Kappenman et al., 2021; Xu et al., 2021). Furthermore, Kroczek et al. (2019) revealed that certain EEG channels are specifically engaged in semantic processing during language comprehension but not directly responsive to visual stimuli, suggesting the need for enriched channel mechanisms for visual semantic modelling. Collectively, these findings strongly indicate that EEG-based visual decoding should incorporate staged and multi-level semantic representation learning rather than relying solely on global embeddings.

Inspired by the above neuroscientific theories, and to address the limitations of existing methods, we propose a brain-mimetic staged representation learning framework for EEG-based visual decoding. Our framework explicitly models three phases that mirror neural visual processing: Phase-I for low-level visual representation learning, Phase-II for high-level semantic representation learning, and Phase-III for integrative information fusion. To further enhance semantic modelling, we introduce two key innovations: (i) a multimodal dual-level semantic learning mechanism that disentangles coarse label-level semantics and fine image-level semantics from visual EEG channels; and (ii) the novel concept of virtual EEG channels, which expand the semantic representational capacity of EEG signals and improve cross-modal alignment.

Extensive experiments on the largest-scale benchmark dataset validate the effectiveness of our framework. Unlike existing methods that rely only on global embedding alignment, our method achieves significant improvements under both subject-dependent and subject-independent zero-shot settings, demonstrating robustness and generalisation across individuals. By bridging neuroscientific theories with advanced representation learning, this work provides a new perspective for EEG-based brain decoding and opens avenues for more biologically grounded, interpretable, and generalisable BCI systems. The main contributions are summarised as follows:

- **Brain-mimetic staged representative learning framework.** We propose a novel brain-decoding paradigm that draws on neuroscientific theories of staged vision. The framework explicitly stages learning into low-level visual perception, high-level semantic abstraction, and integrative information fusion.

- **Multimodal dual-level semantic learning mechanism.** We design a novel semantic learning mechanism that disentangles high-level semantics from EEG visual channels into coarse label-level semantics and fine image-level semantics, capturing richer neural dynamics beyond traditional global embeddings.

- **Virtual EEG channels.** Inspired by neuroscientific evidence of channel specialization, we introduce a novel concept of virtual EEG channels to expand the semantic representational capacity of EEG signals and improve cross-modal alignment. Extensive experiments and ablations validate their effectiveness, demonstrating strong ability to capture coarse semantics in the visual domain.

## 2 RELATED WORK

Recent advances in EEG-based visual decoding have been dominated by efforts to refine EEG encoders in order to obtain stronger embeddings for EEG-vision alignment. Wu et al. (2025) introduced an uncertainty-aware blur prior to suppress noise and improve robustness, while Zhang et al. (2025b) incorporated multimodal priors such as texture and depth to enhance semantic alignment. Zhang et al. (2025a) further leveraged wavelet transforms with contrastive loss to boost category discrimination, and Li et al. (2024) employed guided diffusion models to strengthen EEG–vision correspondence. Other encoder-centric strategies include classifier-based recognition Song et al. (2024), semantic consistency losses Chen et al. (2024), and multimodal graph representations Du et al. (2023), all of which seek to enrich EEG embeddings or reduce modality gaps. Some other studies have extended this encoder-refinement paradigm with external priors or auxiliary supervision. Ferrante et al. (2024a) distilled knowledge from CLIP to inject vision–language semantics into

EEG decoding, Rajabi et al. (2025) proposed human-aligned priors for biologically plausible mappings, and Ma & Ruotsalo (2024) contrasted EEG responses with visual saliency. Others explored 3D spatiotemporal-geometric modelling Xiao et al. (2025), language-guided decoding Song et al. (2025), or entropy-based discriminative training Zeng et al. (2023). Despite these innovations, all existing methods share a fundamental limitation: they treat EEG decoding as a one-shot global embedding alignment problem. As such, they fail to capture the staged dynamics of visual perception and neglect the potential of staged EEG embedding learning and disentangled dual-level semantic modelling.

In contrast, our work introduces a brain-mimetic staged representation learning framework that mirrors the progression of human visual perception from low-level features to high-level semantics and to integrative fusion. We further enrich semantic modelling through a multimodal dual-level disentanglement mechanism and expand EEG's representational capacity with the novel concept of virtual EEG channels. These innovations enable us to more faithfully capture neural dynamics and deliver substantial improvements in both subject-dependent and subject-independent zero-shot decoding.

# 3 PROPOSED BRAIN-MIMETIC DECODING PARADIGM

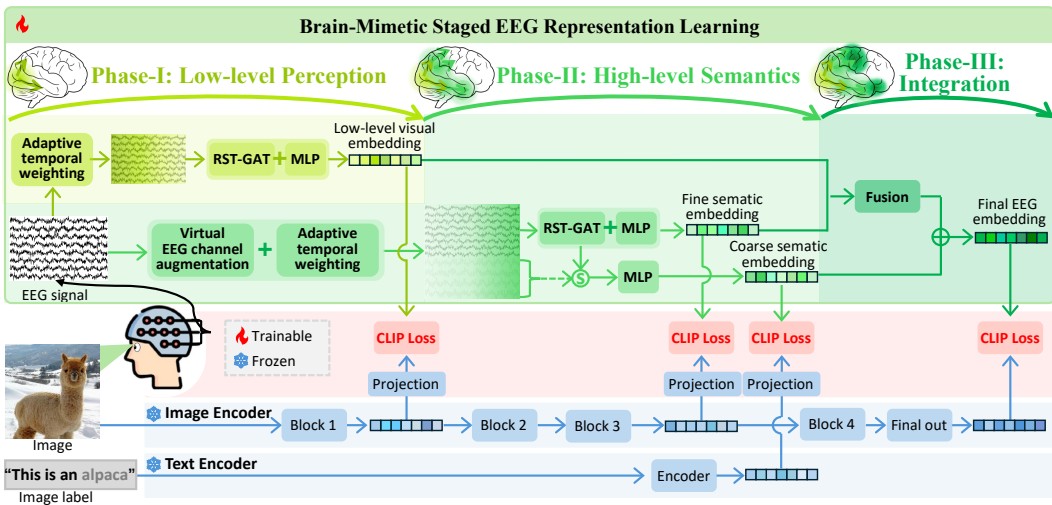

Figure 1: Overview of our proposed brain-mimetic staged EEG representation learning framework. It consists of three phases inspired by neuroscientific theories of progressive and hierarchical vision. **Phase-I** learns low-level visual embeddings from 17 visual-related EEG channels with adaptive temporal weighting and aligns them with image low-level features. **Phase-II** extends the 17 visual channels with virtual EEG channels, applies adaptive temporal weighting, and learns high-level semantics that are disentangled into coarse and fine levels. Coarse semantics capture abstract categorical distinctions and are derived from the virtual channels, while fine semantics reflect more specific and detailed representations and are derived from all visual and virtual channels. Coarse semantics are aligned with text (i.e., image label) features, and fine semantics are aligned with high-level image features. **Phase-III** integrates low-level visual, fine, and coarse semantic embeddings into a unified EEG representation, which is aligned with the final image embeddings. The entire framework is trained with multi-level CLIP losses to ensure consistent cross-modal alignment across all stages.

Most existing methods for EEG-based visual decoding primarily focus on refining EEG encoders to obtain stronger embeddings for alignment with visual features. However, they largely overlook that human visual perception is inherently staged, progressing from early low-level feature detection to higher-level semantic abstraction and ultimately to integrative information fusion. Neuroscientific studies have consistently demonstrated such staged processing in the visual cortex (Fellman & Van Essen, 1991; Goodale & Milner, 1992; Kappenman et al., 2021; Xu et al., 2021). Motivated by these findings, we propose a brain-mimetic decoding framework that explicitly models the three stages of neural visual processing: **Phase-I** for low-level visual representation learning, **Phase-II**

for high-level semantic representation learning, and **Phase-III** for integrative information fusion. To further enrich semantic modelling, we introduce two key innovations: (i) a **multimodal dual-level semantic learning mechanism**, which disentangles coarse label-level semantics and fine image-level semantics from visual EEG channels; and (ii) the new concept of **virtual EEG channels**, which expand the representational capacity of EEG signals and enhance cross-modal alignment.

## 3.1 PHASE-I: LOW-LEVEL VISUAL REPRESENTATION LEARNING

In **Phase-I**, our goal is to learn low-level visual representations from EEG signals and align them with low-level image features. Image features are obtained from the output of `block1` in ResNet50. For EEG, we selectively use 17 channels that are primarily associated with visual responses[1] (Kappenman et al., 2021). In addition, because low-level visual responses are concentrated in early temporal windows (Goodale & Milner, 1992; Graumann et al., 2022), we design a learnable adaptive temporal weighting mechanism to emphasize early EEG signals during representation learning.

Formally, let $(e_v, v)$ denote a batch of EEG-image pairs, where $e_v \in \mathbb{R}^{B \times C_1 \times T}$ represents $B$ EEG samples with $C_1 = 17$ channels and $T$ time steps, and $v \in \mathbb{V}$ denotes the corresponding images. A learnable temporal weight $w_t \in \mathbb{R}^{1 \times T}$ is applied to adaptively emphasize different temporal segments. For low-level EEG visual representation learning, we design a **Residual SpatioTemporal Graph Attention Networks (RST-GAT)**, which incorporates residual connections and spatiotemporal attention into GAT (Veličković et al., 2018) and MLP, to encode the low-level EEG visual representation $e_1 \in \mathbb{R}^{B \times 256}$. This encoding process is defined as:

$$e_1 = f_{\text{MLP}}\Big( f_{\text{ST-GAT}}(e_v \otimes w_t + e_v) + e_v \Big), \tag{1}$$

where $\otimes$ denotes element-wise multiplication with broadcasting along the channel dimension. Meanwhile, the low-level image feature $v_1 \in \mathbb{R}^{B \times 256}$ is obtained by projecting the ResNet50-`block1` output through an MLP. Finally, we employ a CLIP loss to encourage alignment between the low-level EEG embeddings and the corresponding low-level image features:

$$\mathcal{L}_{\text{CLIP}}^I = \text{CLIP\_LOSS}(e_1, v_1). \tag{2}$$

## 3.2 PHASE-II: HIGH-LEVEL SEMANTIC REPRESENTATION LEARNING

**Phase-II** aims to mimic the mid-stage of neural signal processing, where the brain gradually forms and strengthens semantic representations. Our goal is to learn high-level semantic representations from EEG signals. To capture their hierarchical nature, we disentangle them into two levels. **Coarse semantics** capture abstract categorical distinctions and are aligned with text features derived from class labels. **Fine semantics** capture more specific and detailed representations and are aligned with high-level image features. This dual-level design allows the model to reflect both abstract and detailed semantic processing. To support this mechanism, we introduce **virtual EEG channels**. These channels expand the representational capacity of EEG signals and provide a stronger basis for disentangling coarse and fine semantics.

### 3.2.1 VIRTUAL EEG CHANNEL AUGMENTATION

Neuroscientific studies show that a subset of EEG channels (12 in total[2]) are selectively engaged in semantic processing during language comprehension (Kroczek et al., 2019). However, these channels are not responsive to visual stimuli like images. Inspired by this evidence, we propose *virtual EEG channels* to enhance semantic modeling in the visual domain. Analogous to the 12 language-related semantic channels, we construct 12 virtual channels, denoted as $e_{virtual} \in \mathbb{R}^{B \times 12 \times T}$, which are learned and enriched within the proposed *Dual-Level Semantic Learning* module (Section 3.2.2). Ablation experiments (Section 4.5) confirm that these virtual channels improve the EEG representation learning and outperform the real language-related channels, demonstrating their ability to capture semantic information from EEG signals more effectively.

---

[1] 17 channels primarily associated with visual responses: O1, O2, Oz, PO3, PO4, PO7, PO8, POz, P3, P4, P5, P6, P7, P8, Pz, CPz, and Iz (Kappenman et al., 2021).

[2] Semantic-related channels for language comprehension: Fp1, F3, F7, FC5, FC1, C3, T7, CP5, P7, FT7, F5, TP7 (Kroczek et al., 2019; Gifford et al., 2022).

### 3.2.2 MULTIMODAL DUAL-LEVEL SEMANTIC LEARNING MECHANISM

In **Phase-II**, our goal is to model high-level semantics from EEG signals. To better reflect the hierarchical nature of semantic processing, we design a **multimodal dual-level semantic learning mechanism**. The mechanism disentangles EEG-based semantic representations into two levels. **Coarse semantics** capture abstract categorical distinctions, are derived from the virtual channels, and are aligned with text features extracted from image class labels. **Fine semantics** capture specific and detailed representations, are derived from both real and virtual channels, and are aligned with mid- to high-level image features. This multimodal dual-level alignment enables the model to capture richer semantic dynamics beyond global embeddings, thereby enhancing semantic modeling capacity. Since semantic processing in the brain mainly occurs in the mid-to-late temporal stages (Kappenman et al., 2021; Felleman & Van Essen, 1991; Xu et al., 2021), we also design a learnable adaptive temporal weighting mechanism (similar to Phase-I) that emphasizes EEG signals within the corresponding time window.

Formally, after augmentation, the EEG signals are expanded to 29 channels, denoted as $\ddot{e} = [e_v | e_{virtual}] \in \mathbb{R}^{B \times 29 \times T}$, where $e_v \in \mathbb{R}^{B \times 17 \times T}$ corresponds to the 17 visual-related EEG channels (identical to those used in Phase-I, Section 3.1) and $e_{virtual} \in \mathbb{R}^{B \times 12 \times T}$ denotes the proposed 12 virtual EEG channels. An RST-GAT encoder, similar to the one in Phase-I, is deployed to facilitate the encoding process, defined as:

$$\ddot{e}_2 = f_{\text{ST-GAT}}(\ddot{e} \otimes \ddot{w}_t + \ddot{e}) + \ddot{e} = [\ddot{e}_v | \ddot{e}_{virtual}], \tag{3}$$

where $\otimes$ denotes element-wise multiplication with broadcasting along the channel dimension and $\ddot{w}_t \in \mathbb{R}^{1 \times T}$ represents the learnable temporal weight. $\ddot{e}_v \in \mathbb{R}^{B \times 17 \times T}$ corresponds to fine EEG semantics, while $\ddot{e}_{virtual} \in \mathbb{R}^{B \times 12 \times T}$ corresponds to coarse EEG semantics.

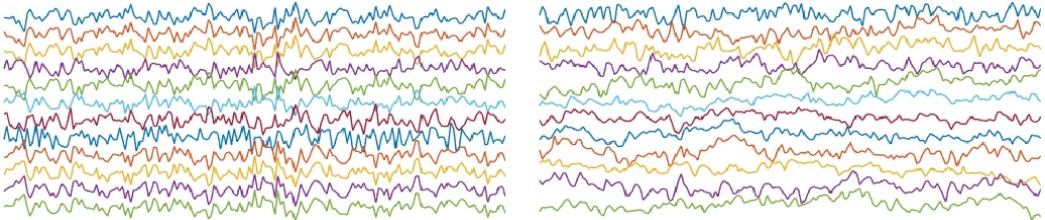

Figure 2: Visual comparison of coarse EEG semantic features: the left panel shows the features ($\ddot{e}_{virtual}$) learned using the proposed 12 virtual EEG channels, while the right panel shows the features learned using 12 real language-related EEG channels in the ablation study **xC-Ori-12** (Section 4.5). The ablation experiments demonstrate that the virtual channels enhance EEG representation learning and achieve superior performance compared to the original language-related channels, obtaining a 4.7 point improvement from 50.3% to 55.0%.

For the coarse EEG semantics, $\ddot{e}_{virtual}$ is first combined with the learnable temporal weight $\ddot{w}_t$ and then passed through an MLP to obtain the coarse semantic representation:

$$\hat{e}_{coarse} = f_{\text{MLP}}(\ddot{e}_{virtual} \circledast \ddot{w}_t), \tag{4}$$

where $\circledast$ denotes weighted temporal aggregation, implemented as batch-wise matrix multiplication along the temporal dimension. The resulting coarse semantic $\hat{e}_{coarse} \in \mathbb{R}^{B \times 1024}$ is aligned with coarse text semantics $t_{coarse} \in \mathbb{R}^{B \times 1024}$ extracted from the pretrained CLIP text encoder (Ilharco et al., 2021). The corresponding alignment loss is:

$$\mathcal{L}_{\text{CLIP-coarse}}^{II} = \text{CLIP\_LOSS}(\hat{e}_{coarse}, t_{coarse}). \tag{5}$$

For the fine EEG semantics, $\ddot{e}_2$ is projected through an MLP to produce the fine semantic representation $\hat{e}_{fine} \in \mathbb{R}^{B \times 1024}$:

$$\hat{e}_{fine} = f_{\text{MLP}}(\ddot{e}_2), \tag{6}$$

which is aligned with fine image semantics $v_{fine} \in \mathbb{R}^{B \times 1024}$ obtained from the projected output of ResNet50-`block3`. The corresponding alignment loss is:

$$\mathcal{L}_{\text{CLIP-fine}}^{II} = \text{CLIP\_LOSS}(\hat{e}_{fine}, v_{fine}). \tag{7}$$

### 3.3 PHASE-III: INTEGRATIVE INFORMATION FUSION

**Phase-III** performs integrative information fusion, where low-level visual features from Phase-I and high-level semantic features from Phase-II are combined into a unified EEG representation. This is inspired by neuroscientific evidence that high-level cognition arises from the progressive integration of multiple information pathways Felleman & Van Essen (1991); Graumann et al. (2022); Goodale & Milner (1992). Specifically, we first fuse the low-level visual feature $e_1$ with the fine semantic representation $\hat{e}_{fine}$, and then integrate the coarse semantic representation $\hat{e}_{coarse}$. The fused EEG representation is finally aligned with the final image embedding extracted by ResNet50.

Formally, the integrative information fusion is computed as:

$$e_{\text{EEG}} = f_{\text{MLP}}([e_1 \,|\, \hat{e}_{fine}]) + \hat{e}_{coarse}, \tag{8}$$

where $[\cdot|\cdot]$ denotes concatenation. The resulting EEG representation $e_{\text{EEG}} \in \mathbb{R}^{B \times 1024}$ is aligned with the final image embedding $v_{\text{image}} \in \mathbb{R}^{B \times 1024}$ through:

$$\mathcal{L}_{\text{CLIP}}^{III} = \text{CLIP\_LOSS}(e_{\text{EEG}}, v_{\text{Image}}). \tag{9}$$

The overall training objective is defined as the weighted sum of the losses from the three phases:

$$\mathcal{L}_{\text{total}} = \alpha_1 \mathcal{L}_{\text{CLIP}}^{I} + \alpha_2 (\mathcal{L}_{\text{CLIP-coarse}}^{II} + \mathcal{L}_{\text{CLIP-fine}}^{II}) + \alpha_3 \mathcal{L}_{\text{CLIP}}^{III}, \tag{10}$$

where $\alpha_1, \alpha_2, \alpha_3$ balance the three phases and are empirically set to 0.1, 0.2, and 0.5, respectively, in our experiments.

## 4 EXPERIMENTS

### 4.1 BENCHMARK DATASET

The benchmark dataset THINGS-EEG (Gifford et al., 2022) was employed for all experiments. THINGS-EEG is currently the largest publicly available EEG dataset for brain decoding, and it has become a widely recognised benchmark in recent top-tier conference and journal publications, particularly in the context of zero-shot learning settings. The dataset was designed to capture rich and generalisable neural representations of visual-semantic concepts, thereby providing a challenging and comprehensive testbed for evaluating model generalisation across subjects and unseen categories. Data were collected using a Rapid Serial Visual Presentation paradigm, in which visual stimuli were presented in rapid succession while EEG signals were recorded. The dataset contains neural responses from 10 participants, each exposed to a broad spectrum of object categories encompassing diverse visual and semantic domains. The **training set** consists of 1,654 object classes, each represented by 10 different images, with each image presented four times in a randomised sequence. This yields a total of 66,160 EEG samples. The **test set** contains 200 held-out classes, each represented by a single image repeated 80 times, resulting in 16,000 EEG samples. All stimuli were presented in a randomised order to reduce habituation and expectancy effects.

Two experimental settings are used in the experiments:
1) **Subject-dependent 200-way zero-shot classification:** The model is trained on the training set and evaluated on the test set of the same subject.
2) **Subject-independent 200-way zero-shot classification:** The model is trained entirely on the training sets of other subjects and evaluated on the test set of the target subject.

### 4.2 BASELINES AND EVALUATION METRICS

We compare our method against seven state-of-the-art EEG visual decoding approaches published in top-tier venues over the past two years, including Wu et al. (2025) (CVPR), Zhang et al. (2025b) (AAAI), Zhang et al. (2025a) (IJCAI), Li et al. (2024) (NeurIPS), Song et al. (2024) (ICLR), Chen et al. (2024) (arXiv), Du et al. (2023) (TPAMI). These methods have advanced the field by introducing priors, multimodal cues, or consistency losses, yet they generally share key limitations: they often emphasise global alignment while overlooking the staged nature of visual processing, and lack mechanisms to integrate low-level, semantic, and fused representations in a biologically informed manner (more discussion in the Related Work, Section 2). Our method addresses these gaps through

a brain-inspired three-phase framework, channel-guided representative learning, and the introduction of virtual channels, enabling more accurate and generalisable EEG-to-vision decoding. Since all of these methods have been evaluated on the same benchmark dataset THINGS-EEG under the two experimental settings and have reported results accordingly, we also follow this protocol and adopt Top-1 and Top-5 classification accuracy as the primary evaluation metrics to ensure consistency and provide a comprehensive assessment of model performance.

### 4.3 KEY IMPLEMENTATION DETAILS

All experiments were carried out on a single NVIDIA GeForce RTX 4090 GPU, with the framework implemented in PyTorch. For EEG signal preprocessing, we follow the standard pipeline described in Song et al. (2024); Wu et al. (2025). For visual feature extraction, we employ the pretrained Open-CLIP ResNet50 model (Ilharco et al., 2021), keeping its parameters fixed throughout training. Model optimization is performed using the AdamW optimizer with a learning rate of 0.0001 and a batch size of 1024. Training is conducted for 40 epochs, with the random seed set to 42 to ensure reproducibility. The same hyperparameter configuration is applied consistently across both subject-dependent and subject-independent experimental settings.

### 4.4 EXPERIMENTAL RESULTS

#### 4.4.1 SUBJECT-DEPENDENT 200-WAY ZERO-SHOT EXPERIMENTS

Table 1 presents the 200-way zero-shot classification results under the subject-dependent setting. Our method achieves the best Top-1 accuracy on all 10 subjects, achieving an average of **55.0%**, which surpasses the strongest prior baseline (Wu et al. (2025) (CVPR), 50.9%) by **+4.1** absolute (**+8.1%** relative). The gains are consistent across subjects, with particularly large margins on Sub01 (+7.3) and Sub09 (+7.4), indicating improved robustness for difficult participants. Compared with recent alignment-based models, e.g., Chen et al. (2024) (arXiv) (37.2%), Zhang et al. (2025a) (IJCAI) (33.4%), Zhang et al. (2025b) (AAAI) (35.6%), and Li et al. (2024) (NeurIPS) (28.5%), our approach improves the average Top-1 accuracy by **17.8-26.5** points, highlighting the effectiveness of staged learning and channel-level modeling beyond global embeddings. For Top-5 accuracy, our method also ranks first with an average of **84.2%**, exceeding Zhang et al. (2025b) (AAAI) (80.2%) and Wu et al. (2025) (CVPR) (79.7%) by **+4.0** and **+4.5** points, respectively (**+5-6%** relative). We obtain the best performance on 8/10 subjects and remain competitive on the remaining two (Sub01 and Sub04), while consistently outperforming Wu et al. (2025) (CVPR) on all subjects.

Table 1: Subject-dependent Top-1 (top) and Top-5 (bottom) accuracy (%) in 200-way zero-shot.

| Methods | Sub01 | Sub02 | Sub03 | Sub04 | Sub05 | Sub06 | Sub07 | Sub08 | Sub09 | Sub10 | Avg. |
|---|---|---|---|---|---|---|---|---|---|---|---|
| Du et al. (2023) (TPAMI) | 6.1 | 4.9 | 5.6 | 5.0 | 4.0 | 6.0 | 6.5 | 8.8 | 4.3 | 7.0 | 5.8 |
| Song et al. (2024) (ICLR) | 13.2 | 13.5 | 14.5 | 20.6 | 10.1 | 16.5 | 17.0 | 22.9 | 15.4 | 17.4 | 16.1 |
| Li et al. (2024) (NeurIPS) | 25.6 | 22.0 | 25.0 | 31.4 | 12.9 | 21.3 | 30.5 | 38.8 | 34.4 | 29.1 | 28.5 |
| Chen et al. (2024) (arXiv) | 32.6 | 34.4 | 38.7 | 39.8 | 29.4 | 34.5 | 34.5 | 49.3 | 39.0 | 39.8 | 37.2 |
| Zhang et al. (2025a) (IJCAI) | 33.0 | 28.0 | 33.5 | 36.0 | 26.0 | 30.5 | 34.0 | 43.0 | 31.5 | 38.5 | 33.4 |
| Zhang et al. (2025b) (AAAI) | 31.4 | 31.4 | 38.2 | 40.4 | 24.4 | 34.8 | 34.7 | 48.1 | 37.4 | 35.6 | 35.6 |
| Wu et al. (2025) (CVPR) | 41.2 | 51.2 | 51.2 | 51.1 | 42.2 | 57.5 | 49.0 | 58.6 | 45.1 | 61.5 | 50.9 |
| Ours | **48.5** | **56.0** | **53.5** | **54.0** | **44.0** | **60.0** | **51.5** | **64.0** | **52.5** | **66.0** | **55.0** |
| Du et al. (2023) (TPAMI) | 17.9 | 14.9 | 17.4 | 15.1 | 13.4 | 18.2 | 20.4 | 23.7 | 14.0 | 19.7 | 17.5 |
| Song et al. (2024) (ICLR) | 39.5 | 40.3 | 42.7 | 52.7 | 31.5 | 44.0 | 42.1 | 56.1 | 41.6 | 45.8 | 43.6 |
| Li et al. (2024) (NeurIPS) | 60.4 | 54.5 | 62.4 | 60.9 | 43.0 | 51.1 | 61.5 | 72.0 | 51.5 | 63.5 | 60.4 |
| Chen et al. (2024) (arXiv) | 63.7 | 69.9 | 73.5 | 72.0 | 58.6 | 68.8 | 68.3 | 79.8 | 69.6 | 75.3 | 69.9 |
| Zhang et al. (2025a) (IJCAI) | 58.5 | 56.5 | 61.0 | 68.0 | 48.0 | 62.5 | 62.5 | 73.5 | 58.5 | 69.0 | 61.8 |
| Zhang et al. (2025b) (AAAI) | **79.7** | 77.8 | 85.7 | **85.8** | 66.3 | 78.8 | 81.0 | 88.6 | 79.4 | 79.3 | 80.2 |
| Wu et al. (2025) (CVPR) | 70.5 | 80.9 | 82.0 | 76.9 | 72.8 | 83.5 | 79.9 | 85.8 | 76.2 | 88.2 | 79.7 |
| Ours | 74.0 | **87.5** | **88.0** | 80.0 | **79.5** | **88.0** | **83.0** | **89.0** | **81.5** | **91.0** | **84.2** |

#### 4.4.2 SUBJECT-INDEPENDENT 200-WAY ZERO-SHOT EXPERIMENTS

Table 2 presents the 200-way zero-shot classification results under the subject-independent setting (**Note**: the methods in Table 1, such as Chen et al. (2024) (arXiv), Zhang et al. (2025a) (IJCAI),

and Zhang et al. (2025b) (AAAI), did not report results for this setting). Our method achieves the best overall Top-1 accuracy of **13.2%**, outperforming the strongest baseline Wu et al. (2025) (CVPR) (12.4%) by **+0.8** points on average. Notably, our framework consistently surpasses all prior approaches on several subjects (e.g., Sub01, Sub02, Sub05, Sub06, and Sub10), and reaches parity with the best baseline on Sub09 and Sub04, demonstrating that the proposed staged learning and channel-level augmentation achieves superior generalization across unseen subjects. Compared with Li et al. (2024) (NeurIPS) (11.8%) and Song et al. (2024) (ICLR) (6.2%), our method improves Top-1 accuracy by **+1.4** and **+7.0** points, respectively, highlighting advances beyond global embedding alignment strategies. For Top-5 accuracy, our method achieves an average of **32.3%**, which is competitive with Li et al. (2024) (NeurIPS) (33.7%) and Wu et al. (2025) (CVPR) (33.4%), and substantially higher than Song et al. (2024) (ICLR) (21.4%) and Du et al. (2023) (TPAMI) (7.0%).

**Discussion.** We note that subject-independent performance is naturally lower than subject-dependent performance, a gap that is well recognized in EEG research. EEG signals are highly identity-dependent, reflecting individual variability in brain anatomy, electrode placement, and cognitive processing, which introduces substantial inter-subject variability (Huang et al., 2023; Saha & Baumert, 2020; Wei & Ding, 2023). Consequently, methods that more faithfully mimic subject-specific neural learning patterns often excel in within-subject decoding but require larger adjustments when generalizing across unseen individuals. In contrast, less biologically grounded approaches may appear less affected across subjects, yet this reflects weaker modeling of true neural dynamics rather than genuine robustness. Despite this inherent challenge, our framework still achieves the best average Top-1 accuracy among all baselines, and consistently delivers stable Top-1 accuracy improvements across most subjects. This robustness confirms that our framework captures genuine neural dynamics, yielding not only superior within-subject decoding but also competitive advantages under the more demanding subject-independent setting.

Table 2: Subject-independent Top-1 (top) and Top-5 (bottom) accuracy (%) in 200-way zero-shot.

| Methods | Sub01 | Sub02 | Sub03 | Sub04 | Sub05 | Sub06 | Sub07 | Sub08 | Sub09 | Sub10 | Avg. |
|---|---|---|---|---|---|---|---|---|---|---|---|
| Du et al. (2023) (TPAMI) | 2.3 | 1.5 | 1.4 | 1.7 | 1.5 | 1.8 | 2.1 | 2.2 | 1.6 | 2.3 | 1.8 |
| Song et al. (2024) (ICLR) | 7.6 | 5.9 | 6.0 | 6.3 | 4.4 | 5.6 | 5.6 | 6.3 | 5.7 | 8.4 | 6.2 |
| Li et al. (2024) (NeurIPS) | 10.5 | 7.1 | **11.9** | **14.7** | 7.0 | 11.1 | **16.1** | **15.0** | 4.9 | **20.5** | 11.8 |
| Wu et al. (2025) (CVPR) | _11.5_ | _15.5_ | _9.8_ | 13.0 | _8.8_ | _11.7_ | _10.2_ | _12.2_ | **15.5** | _16.0_ | _12.4_ |
| Ours | **13.0** | **16.2** | 8.0 | _14.5_ | **10.0** | **14.0** | 9.5 | 11.5 | _14.5_ | **20.5** | **13.2** |
| Du et al. (2023) (TPAMI) | 8.0 | 6.3 | 5.9 | 6.7 | 5.6 | 7.2 | 8.1 | 7.6 | 6.4 | 8.5 | 7.0 |
| Song et al. (2024) (ICLR) | 22.8 | 20.5 | 22.3 | 20.7 | 18.3 | 22.2 | 19.7 | 22.0 | 17.6 | 28.3 | 21.4 |
| Li et al. (2024) (NeurIPS) | 26.8 | 24.8 | **33.8** | **39.4** | 23.9 | **35.8** | **43.5** | **40.3** | 22.7 | **46.5** | **33.7** |
| Wu et al. (2025) (CVPR) | _29.7_ | _40.0_ | _27.0_ | 32.3 | **33.8** | 31.0 | 23.8 | _32.2_ | **40.5** | _43.5_ | _33.4_ |
| Ours | **32.0** | **41.5** | 22.0 | _34.5_ | _31.5_ | _31.5_ | 27.0 | 30.5 | _32.0_ | 40.0 | 32.3 |

## 4.5 ABLATION STUDY

To rigorously validate the rationale behind each design of our framework, we conduct the following ablation experiments:

**(0) Ours-All**: The full version of our proposed method with all components enabled.

**(1) xC-Ori-12**: The proposed 12 virtual channels (i.e., $e_{virtual}$ in Section 3.2.1) are removed and replaced with the 12 real language-related semantic EEG channels.

**(2) xC-Ori-12-xC**: Based on ablation experiment **xC-Ori-12**, the coarse semantic branch is further disabled (i.e., Eq. (4) and Eq. (5) are removed).

**(3) xC-x12V**: The 12 virtual channels are removed; only the 17 visual EEG channels are used, and the coarse semantic branch (Eq. (4)) is computed using features from all 17 channels.

**(4) xP-xPhaseI**: Phase-I is disabled.

**(5) xP-xPhaseII**: Phase-II is disabled.

**(6) xP-PhaseII-xF**: The fine semantic branch (Eq. (6) and Eq. (7)) in Phase-II is disabled.

**(7) xP-PhaseII-xC**: The coarse semantic branch (Eq. (4) and Eq. (5)) in Phase-II is disabled.

We conduct ablation studies in the subject-dependent 200-way zero-shot setting. Table 3 summarises the contribution of each component. The full model (**Ours-All**) attains the highest Top-

1/Top-5 averages of **55.0%/84.2%**. Replacing the proposed *virtual EEG channels* with the real 12 language-related semantic channels (**xC-Ori-12**) reduces Top-1 to **50.3%** (-4.7) and Top-5 to **82.4%** (-1.8), indicating that virtual channels provide a stronger semantic carrier for visual EEG. Removing the coarse semantic branch on top of this replacement (**xC-Ori-12-xC**) further drops performance to **49.0%/80.6%** (Top-1/Top-5), showing that coarse label-level supervision contributes to robustness beyond fine-grained cues. When virtual channels are removed but only the 17 visual channels are kept (**xC-x12V**), the model performs reasonably (**53.4%/83.0%**), yet lags behind **Ours-All** by 1.6/1.2 points (Top-1/Top-5), confirming the net gain brought by virtual-channel augmentation.

Eliminating the low-level phase (**xP-xPhaseI**) leads to **52.4%/81.8%**, evidencing the necessity of early-stage low-level alignment for stabilizing mid-/late-stage learning. Removing the dual-level semantic phase altogether (**xP-xPhaseII**) causes a larger degradation to **48.7%/79.3%**, underscoring the central role of Phase-II. Within Phase-II, disabling the fine semantic branch (**xP-PhaseII-xF**) yields **47.8%/80.4%**, whereas disabling the coarse semantic branch (**xP-PhaseII-xC**) gives **53.0%/83.3%**. Thus, the *fine semantic* branch is the primary driver for Top-1 discrimination (largest drop when removed), while the *coarse semantic* branch improves calibration/recall (clear Top-5 gain), and their combination with Phase-I features (Phase-III fusion) delivers the best overall accuracy and consistency across subjects.

Overall, these results validate that (1) Phase-I anchors early low-level signals, (2) Phase-II's dual-level semantics together with virtual channels capture richer neural dynamics, and (3) Phase-III integrates them into a unified representation, jointly achieving state-of-the-art performance.

Table 3: Ablation studies on subject-dependent Top-1 (top) and Top-5 (bottom) accuracy (%).

| Methods | Sub01 | Sub02 | Sub03 | Sub04 | Sub05 | Sub06 | Sub07 | Sub08 | Sub09 | Sub10 | Avg. |
|---|---|---|---|---|---|---|---|---|---|---|---|
| Ours-All | 48.5 | **56.0** | **53.5** | **54.0** | 44.0 | **60.0** | **51.5** | 64.0 | **52.5** | **66.0** | **55.0** |
| xC-Ori-12 | 45.5 | 51.5 | 46.5 | 51.0 | 42.5 | 55.5 | 42.0 | 61.5 | 47.0 | 59.5 | 50.3 |
| xC-Ori-12-xC | 45.0 | 52.0 | 47.5 | 52.5 | 41.5 | 54.5 | 42.0 | 61.5 | 45.5 | 62.0 | 50.4 |
| xC-x12V | 45.0 | 53.5 | 51.0 | 52.5 | **45.5** | 57.0 | 50.5 | **67.0** | 51.0 | 61.0 | 53.4 |
| xP-xPhaseI | 46.5 | 54.5 | 51.0 | 50.0 | **45.5** | 56.5 | 48.5 | 64.5 | 49.0 | 63.5 | 53.0 |
| xP-xPhaseII | 46.0 | 47.0 | 50.5 | 50.5 | 42.0 | 54.5 | 47.5 | 56.5 | 42.5 | 55.0 | 49.2 |
| xP-PhaseII-xF | 49.0 | 44.0 | 46.5 | 49.0 | 38.0 | 55.0 | 46.5 | 57.5 | 42.0 | 57.0 | 48.5 |
| xP-PhaseII-xC | **51.0** | 55.0 | 51.0 | 49.5 | **45.5** | 59.0 | 50.5 | 63.5 | 49.5 | **66.5** | 54.1 |
| Ours-All | 74.0 | **87.5** | **88.0** | 80.0 | **79.5** | **88.0** | 83.0 | 89.0 | 81.5 | **91.0** | 84.2 |
| xC-Ori-12 | 76.0 | 81.0 | 83.0 | **83.5** | 74.0 | 81.5 | 78.0 | 88.0 | 80.5 | 87.5 | 81.3 |
| xC-Ori-12-xC | 75.5 | 82.5 | 84.0 | 83.0 | 76.5 | 82.0 | 77.5 | 88.0 | 80.5 | 89.0 | 81.9 |
| xC-x12V | 75.5 | 84.5 | 87.5 | 80.5 | 77.0 | 85.0 | 81.0 | 89.0 | 83.5 | 88.0 | 83.2 |
| xP-xPhaseI | 72.5 | 85.0 | 85.0 | 78.5 | 77.0 | 86.5 | 80.5 | 89.0 | 83.5 | 88.5 | 82.6 |
| xP-xPhaseII | 78.0 | 82.5 | 81.0 | 79.0 | 74.5 | 83.5 | 78.0 | 85.0 | 79.0 | 86.0 | 80.7 |
| xP-PhaseII-xF | 76.0 | 81.5 | 82.5 | 82.5 | 71.5 | 83.5 | 76.5 | 85.0 | 79.0 | 86.0 | 80.4 |
| xP-PhaseII-xC | 73.5 | 84.5 | 87.0 | 80.5 | 78.5 | 86.0 | 82.5 | 90.0 | 80.0 | 90.5 | 83.3 |

## 5 CONCLUSION

This work establishes a new paradigm for EEG-based visual decoding by grounding EEG representation learning in the staged principles of human visual perception. Rather than treating EEG decoding as a single-step global alignment task, our brain-mimetic framework demonstrates how low-level perception, hierarchical semantic abstraction, and integrative fusion can be explicitly modeled within a unified system. Through the introduction of dual-level multimodal semantic learning and virtual EEG channels, we extend the representational capacity of EEG signals and show how biologically inspired design can translate into measurable improvements in robustness and generalization. Experiments and ablations on large-scale benchmarks confirm that this paradigm consistently advances the frontier of EEG-based visual decoding. Beyond its immediate performance gains, our study highlights the promise of bridging neuroscience and machine learning. By aligning computational models with staged neural processes, we open new perspectives for building more reliable and generalizable brain–computer interfaces and for advancing brain-inspired artificial intelligence. Looking ahead, further work may explore adaptive strategies to better mitigate inter-subject variability, extending the reach of this paradigm to broader real-world applications.

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

# A APPENDIX

This manuscript is the authors' original work. Except for minor English grammar checking with ChatGPT, no large language model or AI tool was used for idea generation, problem formulation, literature search or screening, methodology design, code implementation, data processing, experimental design, statistical analysis, figure or table drafting, or substantive writing. All intellectual contributions, including conceptualization, model design, and empirical evaluation, are solely those of the authors.

