# OpenReview forum: "Brain-Mimetic Staged Representation Learning with Disentangled Coarse and Fine Semantic for EEG Visual Decoding"
_ICLR.cc/2026/Conference — Submitted to ICLR 2026_

### Official Review · Reviewer_MERQ · 2025-11-01

**Soundness:** 3
**Presentation:** 4
**Contribution:** 3
**Rating:** 4
**Confidence:** 3

**Summary:**

The study pioneers a biologically grounded EEG visual decoding method, bridging neuroscience and multimodal machine learning. By explicitly modeling staged perception and disentangled semantics, it achieves robust decoding performance and opens new directions for brain-inspired AI and generalizable BCI systems.

**Strengths:**

- The study draws on neuroscientific insights and proposes a three-phase learning process: low-level visual perception, high-level semantic abstraction, and integrative fusion. This makes the model more grounded in how human vision.

- The study innovatively proposes “virtual EEG channels” as a way to expand the representational capacity of the EEG signal space and improve semantic alignment in the visual-EEG domain. This enhances the cross-modal alignment between EEG and vision/text features.

- On a public benchmark dataset, the method outperforms previous state-of-the-art techniques in both subject-dependent and subject-independent zero-shot settings. This demonstrates that the approach is not only accurate but also more generalizable across different subjects.

**Weaknesses:**

- Although the authors used a publicly available dataset (THINGS‑EEG) for evaluation, it only includes a relatively small number of participants (10 subjects). As a result, the model’s cross-subject generalization may still be constrained by the limited sample diversity. Maybe more datasets should be tested on.

- The ablation experiments demonstrate performance gains brought by the virtual EEG channels. However, in-depth analysis is needed to explain the exact neurophysiological correspondence of these virtual channels. If their role is merely to increase the size of the representational space, the accuracy of using the term "channels" should be questioned.

- The paper only includes one ablation study and lacks in-depth analysis related to neuroscientific priors. For instance, it fails to explore the effectiveness of EEG representations at different levels and whether these representations can truly reflect coarse-grained and fine-grained semantics. The authors could address this gap through experiments such as reconstructing images from EEG representations at different levels or conducting analyses on how these representations contribute to specific objects that rely more on either coarse-grained or fine-grained semantics.

**Questions:**

- Why does the ablation study only conduct in the subject-dependent 200-way zero-shot setting?

---

### Official Review · Reviewer_314k · 2025-11-01

**Soundness:** 3
**Presentation:** 3
**Contribution:** 3
**Rating:** 4
**Confidence:** 3

**Summary:**

The paper proposes a brain-inspired three-stage framework for EEG→vision zero-shot decoding. Stage I learns low-level visual embeddings from vision-related EEG channels and aligns them to early image-backbone features (CLIP-style). Stage II introduces learnable “virtual channels,” splits semantics into coarse (label/text-aligned) and fine (image-feature-aligned) branches, and applies adaptive temporal weighting. Stage III fuses the three streams and aligns the fused EEG embedding to the final image embedding via multi-level contrastive losses. Experiments are limited to THINGS-EEG under two 200-way settings (subject-dependent / subject-independent) with Top-1/Top-5 and ablations over stages/branches/virtual channels.

**Strengths:**

1．The idea is interesting: The paper explains why it moves from “one-shot global alignment” to a staged “low-level → semantic → fusion” pipeline. This mapping fits the EEG/BCI view that prior knowledge about human visual processing should be leveraged. As a structural idea, it is novel and meaningful for the field.

2．Multi-level alignment & temporal weighting: The combination of CLIP-style multi-level alignment, temporal weighting, and a sensible EEG encoder is coherent; ablations reasonably attribute gains to modules.

3．Subject-dependent performance: Clear improvements in the subject-dependent setting; the subject-independent gains are smaller but directionally consistent.

4．Reusable recipe: The paper offers a concrete “how-to” for EEG×vision alignment that later work can adapt (swap encoders, modify semantic branches, etc.).

**Weaknesses:**

1．External validity: Evidence comes from a single dataset and paradigm. No cross-dataset or cross-paradigm checks (imagery/video, different presentation rates), and no tests across layouts/devices.

2．Dataset–claim fit: THINGS-EEG states “each image was presented for 50 ms.” For a claim centered on high-level semantic alignment, 50 ms is short; the paper needs stronger justification or complementary evidence.

3．Virtual channels: scientific positioning: They boost performance but are not a classical neuro prior. Without spatial–temporal characterization and statistical links to language-related channels, the “brain-inspired” narrative is weakened.

4．Stability & statistics: Results look like single-seed estimates; there’s no ≥5-seed mean±std/CI, and no paired permutation/nonparametric tests with effect sizes vs. the strongest baselines.

5．Interpretability vs. Claims: If “staging + semantic disentanglement” is the core claim, we should see channel/region importance maps, temporal weight curves vs. canonical ERP windows, and RSA/CKA/CCA mapping of stages/branches to backbone layers (incl. the text branch).

6．Subject-independent comparisons are not fully symmetric (some prior work lacks that setting); which baselines are reproduced vs. quoted isn’t always clear. Aligning to CLIP features is now standard—the contribution reads as systems-level structuring rather than a new learning objective.

7．Subject-independent failure modes: Performance is comparatively weaker; attributing it solely to identity-dependence of EEG is plausible but incomplete without an analysis of this model’s (and baselines’) failure patterns.

**Questions:**

1．Address the 50 ms presentation head-on: THINGS-EEG explicitly says “each image was presented for 50 ms.” To sustain a high-level semantics claim under such a short window, either provide direct evidence (e.g., temporal weight curves aligning with known ERP latencies; representational similarity to higher backbone layers) or add results with longer presentations / different paradigms (imagery/video) as a counterpoint.

2．Cross-setting validation: Add a light but complementary experiment (second dataset, imagery/video paradigm, or varied presentation rates) to demonstrate generalization beyond THINGS-EEG.

3．Stability & significance: Report ≥5-seed mean±std (or CI) and run paired permutation/nonparametric tests with effect sizes against the strongest baseline; show error bars in plots.

4．Interpretability aligned to the claim: Provide channel/region topographies, temporal weighting vs. ERP windows, and RSA/CKA/CCA mapping between stages/branches and backbone layers (incl. the text branch) to ground the “staging + semantic disentanglement” story.

5．Profile the virtual channels: Show spatial topographies and temporal heatmaps, relate them to language-related channels, and include statistical tests. Also add a systematic full-pipeline ± virtual channels comparison to separate physiologically plausible gains from pure engineering augmentation.

6．Baseline/setup transparency: In the subject-independent setting, enforce symmetric configurations; clearly mark which numbers are reproduced vs. quoted and unify evaluation protocols and hyperparameter ranges.

---

### Official Review · Reviewer_GrfV · 2025-11-02

**Soundness:** 2
**Presentation:** 3
**Contribution:** 2
**Rating:** 2
**Confidence:** 4

**Summary:**

This paper tackles the challenge of decoding visual information from EEG signals by proposing a novel framework inspired by the human brain's visual processing pathway. Instead of focusing solely on the EEG encoder, the authors model visual perception as a three-stage process, which seems to be the main contribution.

**Strengths:**

The model's three-stage architecture (low-level features, high-level semantics, fusion) is directly inspired by the human visual processing pathway.

The paper introduces dual-level semantic learning (disentangling coarse and fine semantics) and virtual EEG channels, which are solutions to enhance the richness of the EEG representation.

**Weaknesses:**

Implementing a multi-stage framework that incorporates several innovative components, such as virtual channels, may be considerably more complex than employing a straightforward, end-to-end encoder system. This added complexity can pose challenges in terms of implementation, training, and interpretation, potentially making it more difficult for researchers and practitioners to derive meaningful insights from the results.

The introduction of virtual EEG channels appears to serve as a computational construct designed to enhance signal processing capabilities. However, the relationship between these virtual channels and their direct neuroscientific implications remains ambiguous. There may be ongoing discourse regarding whether their primary role is to serve as a legitimate representation of neural activity or merely as a sophisticated data augmentation technique that boosts the overall performance of the model.

Furthermore, while this framework is informed by principles drawn from human vision, there is no assurance that adhering to this prescribed three-stage process will yield the most effective outcomes for a machine learning model. It is quite plausible that an unconstrained model might discover alternative, more efficient representations of the data, leading to improved performance and potentially offering novel insights that the rigid framework does not capture.

**Questions:**

See Weaknesses.

**Details Of Ethics Concerns:**

nil

---

### Official Review · Reviewer_5y4h · 2025-11-02

**Soundness:** 2
**Presentation:** 4
**Contribution:** 3
**Rating:** 4
**Confidence:** 4

**Summary:**

This paper proposes a brain-mimetic staged representation learning framework for EEG-based visual decoding, inspired by hierarchical human visual perception. Specifically, the proposed framework has three stages: (1) low-level visual representation learning from visual EEG channels; (2) high-level semantic representation learning via a multimodal dual-level semantic disentanglement (coarse-level and fine-level); (3) integrative fusion for unified EEG embeddings. The system is trained with multi-level CLIP losses across phases. Experiments show impressive improvements over prior state-of-the-art methods. Ablation studies validate the contribution of each proposed component.

**Strengths:**

* The paper is well motivated by neuroscience. The proposed staged framework is well connected to low-level and high-level visual cognition.
* The proposed multimodal dual-level semantic learning mechanism is reasonable and effective. By disentangling coarse (text-level) and fine (image-level) semantics and utilizing CLIP semantic features, the framework models EEG-vision understanding in a structured and hierarchical way.
* Experimental results demonstrate decent performance gains brought by the proposed model. The ablations are also thorough.

**Weaknesses:**

* About the virtual EEG channels:
    * It's not very clear how the virtual EEG channels are obtained. Are they initialized from scratch as learnable vectors? or obtained from previous EEG signals? If initialized from scratch, how can this be related to visual stimuli in cognitive science?
    * The channel number is set to 12 in an arbitrary way without any ablation.
* Missing experiments with other datasets: The THINGS-EEG dataset is the only benchmark used for both training and evaluation. To validate the generalizability of the proposed method, it is suggested to experiment with other datasets as well.
* Missing detailed comparison. Compared with other baselines, the proposed model employs additional CLIP modules and multi-phase modules. However, the detailed comparison with previous works regarding model parameters and inference FLOPs is missing.
* In Tab2, the subject-independent top-5 accuracy of proposed method is worse than some of other competitors while top-1 accuracy is outstanding. Can author(s) explain why?

**Questions:**

Refer to weaknesses

---

### Meta-Review · Area_Chair_mWbV · 2026-01-05

**Summary:**

The paper proposes a three-stage framework for neuroscience-inspired EEG visual decoding.

Strength: The reviewers acknowledge that the work is well-motivated by neuroscience. They also find some novel ideas.

Weakness: Weaknesses that the reviewers identify are quite consistent.
(1) The method is validated only one, rather small dataset, so it is questionable whether the method can generalize well.
(2) The idea of 'virtual channels' is interesting but its connection to neuroscientific implications and its effects are not clearly presented. In-depth analysis and ablation for justifying effectiveness of the proposed method are also lacking.
(3) The performance of the subject-independent case is rather weak.

**Reviewer Concerns:**

No rebuttal was submitted.

**Reviewer Scores:**

The reviewers' concerns are consistent, and no rebuttal was submitted. It is unlikely that the reviewers would have changed their scores.

---

### Decision · Program_Chairs · 2026-01-26

Reject